

# Identification of a five-gene signature in association with overall survival for hepatocellular carcinoma

Lei Yang, Weilong Yin, Xuechen Liu, Fangcun Li, Li Ma, Dong Wang and Hongxing Li

Department of Histology and Embryology, Binzhou Medical University, Yantai, Shandong, China

## ABSTRACT

**Background**. Hepatocellular carcinoma (HCC) is considered to be a malignant tumor with a high incidence and a high mortality. Accurate prognostic models are urgently needed. The present study was aimed at screening the critical genes for prognosis of HCC.

**Methods**. The GSE25097, GSE14520, GSE36376 and GSE76427 datasets were obtained from Gene Expression Omnibus (GEO). We used GEO2R to screen differentially expressed genes (DEGs). A protein-protein interaction network of the DEGs was constructed by Cytoscape in order to find hub genes by module analysis. The Metascape was performed to discover biological functions and pathway enrichment of DEGs. MCODE components were calculated to construct a module complex of DEGs. Then, gene set enrichment analysis (GSEA) was used for gene enrichment analysis. ONCOMINE was employed to assess the mRNA expression levels of key genes in HCC, and the survival analysis was conducted using the array from The Cancer Genome Atlas (TCGA) of HCC. Then, the LASSO Cox regression model was performed to establish and identify the prognostic gene signature. We validated the prognostic value of the gene signature in the TCGA cohort.

**Results**. We screened out 10 hub genes which were all up-regulated in HCC tissue. They mainly enrich in mitotic cell cycle process. The GSEA results showed that these data sets had good enrichment score and significance in the cell cycle pathway. Each candidate gene may be an indicator of prognostic factors in the development of HCC. However, hub genes expression was weekly associated with overall survival in HCC patients. LASSO Cox regression analysis validated a five-gene signature (including CDC20, CCNB2, NCAPG, ASPM and NUSAP1). These results suggest that five-gene signature model may provide clues for clinical prognostic biomarker of HCC.

Corresponding authors
Dong Wang, wangdby@163.com
Hongxing Li, bylihx@163.com

## INTRODUCTION

The incidence rate of hepatocellular carcinoma (HCC) ranks sixth among all malignant tumors and the mortality rate ranks third (*Bray et al., 2018*). More than 580,000 new cases are expected in Asia every year (*Siegel, Miller & Jemal, 2019*). The genetic aberrations,

cellular environment and environmental effects are considered as responsible for the development, progression and metastasis of HCC (*Bray et al., 2018*).

Genomic research has been the focus of hepatocellular carcinoma treatment (*Yan et al., 2019*). Recently, high-throughput platform microarrays for analyzing gene expression have been widely developed (*Mari et al., 2019*) as an effective tool for identifying general genetic changes during tumorigenesis (*Wan et al., 2019*). Microarray techniques can not only find related genes of diseases, targets of anti-tumor drugs, but also prognostic analysis of tumor patients, and can reveal the relationship between gene expression and regulation. In clinical research, they also play the role of providing ideas for the diagnosis and treatment of certain diseases (*Szuhai & Vermeer, 2015*). We found that there have been studies exploring the prognostic signatures of colon cancer and lung adenocarcinoma, but the prognostic signatures of hepatocellular carcinoma need to be supplemented (*Cao et al., 2020*; *Wei et al., 2018*).

In this study, we chose four GEO series (GSE25097, GSE14520, GSE36376 and GSE76427) including hepatocellular carcinoma tumor tissue and non-tumor tissue samples. We use GEO2R to screen out differentially expressed genes (DEGs) and to find hub genes by constructing their protein interaction network. This study is aimed at validating some potential targets to effectively assist clinical workers to predict overall survival of HCC patients.

## MATERIALS & METHODS

### Data adoption criteria

We have looked for publicly available series from the GEO Repository browser. Using "hepatocellular carcinoma" as a keyword, a total of 450 series were retrieved. Sort these series acting in accordance with the number of samples, set study type to "Expression profiling by array" and organism to "Homo sapiens", and look for the series with normal tissue and hepatocellular carcinoma tissue control. Make sure that the normal samples are taken from adjacent tissues of HCC patients. In the end, four HCC gene expression profiles (GSE14520, GSE25097, GSE36376 and GSE76427) were selected because they have more and better-quality samples. The normal samples in these four series were all taken from adjacent tissues of HCC patients. Data were downloaded from the publicly available database hence it was not applicable for additional ethical approval.

### DEGs analysis

GEO2R (http://www.ncbi.nlm.nih.gov/geo/geo2r/) was implemented to screen out DEGs between HCC tumor and non-tumor tissue samples. After we obtain the data, we used Bio Tools v5.0 (http://www.chrisapp.xyz:3838/R/AnnoE2/) to draw a volcano map to find statistically significant differentially expressed genes. The adjusted $P < 0.01$ and $|logFC| \geq 1$ were set as the threshold, so the false positive result was eliminated as much as possible. In order to eliminate the background error caused by different research units on different platforms, we use the Venny 2.1.0 (https://bioinfogp.cnb.csic.es/tools/venny) mapping to screen out the shared DEGs.

## PPI network construction

We imported initially screened genes into the STRING database (http://string-db.org/), web-based software designed to calculate the integration of protein-protein interactions (*Ashburner et al., 2000*) to obtain the highest confident genes (0.900). Then, the MCC algorithm in Cytohubba which an APP in Cytoscape (Version: 3.7.2) was utilized to screen out the hub genes (*Chin et al., 2014*).

## Functional and pathway enrichment analysis

Metascape (http://metascape.org) was performed to analyze process enrichment analysis and pathway analysis of neighbor genes of hub genes. On the basis of Metascape tool, The Gene Ontology (GO) terms, Kyoto Encyclopedia of Genes and Genomes (KEGG) pathways and Reactome Gene Sets can be analyzed. Terms with a *P* value<0.01, minimum count of 3, and an enrichment factor >1.5 are collected and grouped into clusters depended on their membership similarities. Moreover, the MCODE algorithm of the network is used to identify closely connected neighborhoods of proteins (*The Gene Ontology Consortium, 2019*; *Kanehisa et al., 2017*; *Wang et al., 2013*).

## Gene Set Enrichment Analysis (GSEA) analysis

GSEA (v4.0.3) was used to verify the results of Metascape analysis (*Subramanian et al., 2007*; *Mardinoglu, Gatto & Nielsen, 2013*). The gene sets file (C2 KEGG v7.0 symbols), phenotype labels file, and expression dataset file and chip annotation file were prepared and loaded into GSEA. $P < 0.05$ was considered statistically significant.

## The hub genes' transcription level analysis in patients with HCC

ONCOMINE (http://www.oncomine.org) was performed to analyze the mRNA levels of hub genes in HCC (*Rhodes et al., 2004*; *Rhodes et al., 2007*). Threshold limits were as follows: the data type was mRNA, fold change = 2 and $P = 0.01$. Differential analysis between normal and tumor tissues was performed for each gene. Meanwhile, we compared the level of hub genes expression between normal and tumor tissues by Gene Expression Profiling Interactive Analysis (GEPIA 2) (*Tang et al., 2019*). We set |Log2FC| Cutoff to 1, Jitter Size to 0.4, *P*-value Cutoff to 0.01, and compared HCC tumor samples ($n = 369$) from the TCGA database with normal samples ($n = 160$) from the TCGA databases.

## Survival analysis of hub genes

The Kaplan–Meier plotter (http://www.kmplot.com) contains gene expression data from 364 clinical HCC patients derived from TCGA (*Menyhárt, Nagy & Győrffy, 2018*). According to the median expression, these samples were divided into a low expression group and a high expression group. The Kaplan–Meier plotters were used to calculate the relapse-free survival (RFS), progression-free survival (PFS) and overall survival (OS) of all liver cancer patient samples.

## Establishment of the prognostic gene signature

The mRNA expression and clinical data were downloaded from TCGA-LIHC and cBioportal, including 374 TCGA-LIHC and 50 normal control samples. All patients with a follow-up period less than 60 days were excluded for survival analysis. A prognostic
gene signature was constructed based on the results of the least absolute shrinkage and selection operator (LASSO). Cox regression model coefficients ($\beta$) multiplied with its mRNA expression level. The risk score = ($\beta$gene 1 * expression level of gene 1) + ($\beta$gene 2 * expression level of gene 2) + ($\beta$gene 3 * expression level of gene 3) + ...+ ($\beta$gene n * expression level of gene n) (*Huitzil-Melendez et al., 2010*). We used the Survminer R package to find the optimal cut-off values. Then the Kaplan–Meier survival curve combined with a log-rank test was performed to compare the difference in overall survival between the high-risk score group and low-risk score group.

## Statistical analysis

GraphPad Prism version 8.0 and R software version 4.0.2 (GraphPad Software Inc., USA) was used for statistical analyses. All tests were two-sided, $P < 0.05$ was considered statistically significant.

# RESULTS

## Identification of DEGs

We did the research as described in the flow chart (Fig. 1). In order to screen the difference of gene expression between HCC and normal liver tissue, four gene expression series (GSE14520, GSE25097, GSE36376 and GSE76427) were downloaded from the GEO database. The profile of GSE14502 includes 225 HCC tumor tissues and 220 non-tumor tissues. The profile of GSE25097 includes 268 HCC tumor tissues and 243 non-tumor tissues. The profile of GSE36376 includes 240 HCC tumor tissues and 193 non-tumor tissues. The profile of GSE76427 includes 115 HCC tumor tissues and 52 non-tumor tissues (Table 1). We found 1095 DEGs in GSE14520 (Fig. 2A), 1872 DEGs in GSE25097 (Fig. 2B), 688 DEGs in GSE36376 (Fig. 2C) and 488 DEGs in GSE76427 (Fig. 2D). Among them, 142 DEGs were detected in all four datasets (Fig. 2E) and all their expressions were matched, including 26 up-regulated genes (Fig. 2F) and 116 down-regulated genes (Fig. 2G) in HCC tumor tissue samples compared with non-tumor liver tissue samples.

## PPI network construction and co-expression analysis in patients with HCC

Next, we imported 142 genes into the STRING webpage software for PPI network construction. After setting the minimum required interaction score to highest confidence (0.900), we obtained 121 nodes and 398 edges. Then, a cluster network was created by using the MCL cluster algorithm in the STRING website. The first cluster included 16 genes (AURKA, CDKN3, CCNB2, CDC20, PTTG1, MELK, RACGAP1, PRC1, TOP2A, NUSAP1, ASPM, NCAPG, RFC4, PHLDA1, MCM6 and MCM2 (Fig. 3A). Next, we applied Cytohubba's MCC algorithm to rank and obtained the top 10 central genes. They were CDC20, CCNB2, AURKA, ASPM, NCAPG, NUSAP1, CDKN3, PRC1, MELK and TOP2A (Fig. 3B). Interestingly, these ten genes were all within the first cluster created by the MCL clustering algorithm.

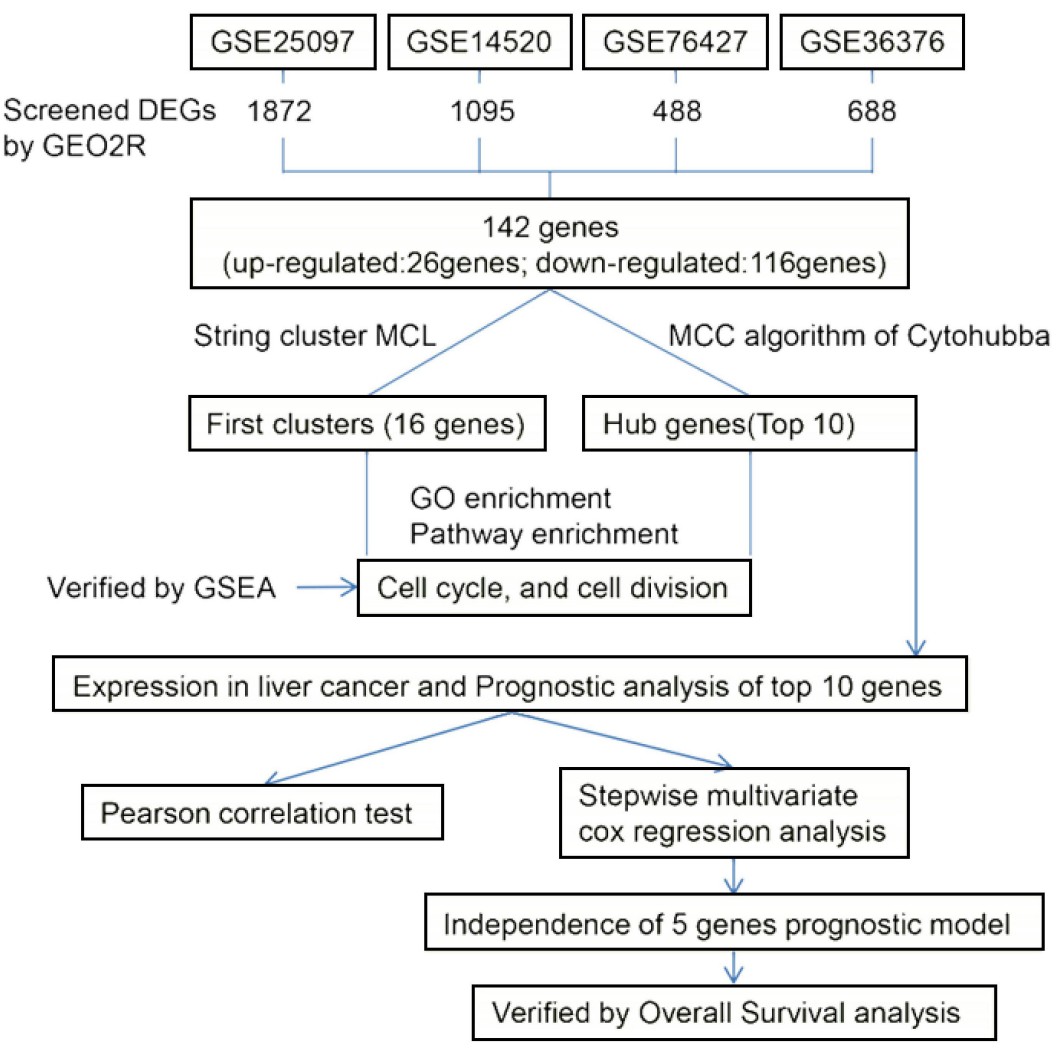

**Figure 1** The flow chart showing our protocol for studying the mRNA prognostic characteristics of HCC.

## Functional and pathway enrichment analyses

We used the Benjamini and Yekutieli method to adjust the $P$ value of the enrichment analysis results, and used the standard of adj. $P < 0.05$ to screen for significant enrichment regions. GO terminology enrichment analysis indicated that 10 genes were mainly enriched in cell division, mitotic cell cycle process, DNA conformation change, positive regulation of apoptotic process, DNA-dependent ATPase activity, spindle pole and neural precursor cell proliferation. On the other side, KEGG pathway and Reactome gene sets based analysis revealed that the genes were mainly enriched in the cell cycle (Figs. 4A–4C). We used PPI network construction in Metascape, and extracted the most important MCODE components from it, and performed functional and pathway enrichment analysis for each MCODE component. The results indicated that the candidate genes of the cell cycle pathway may be indicators of prognostic factors in patients with HCC (Fig. 4D). In order to validate

| Table 1 | Basic information of four GEO datasets. | | |
|---|---|---|---|
| **GEO datasets** | **Platform** | **Hepatocellular carcinoma samples** | **Non-tumor samples** |
| GSE25097 | GPL16087 | 243 | 268 |
| GSE47197 | GPL16699 | 61 | 63 |
| GSE54236 | GPL6480 | 80 | 81 |
| GSE60502 | GPL96 | 18 | 18 |

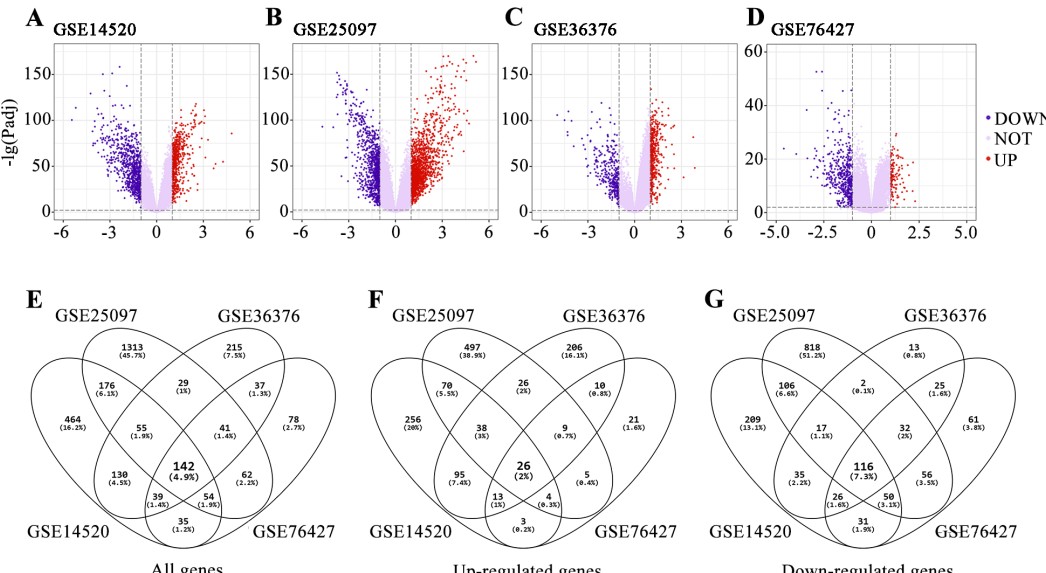

**Figure 2** **Screening of Commonly Differentially Expressed Genes in the four hepatocellular carcinoma datasets.** (A) DEGs in GSE14520 are displayed in the volcano plot. (B) DEGs in GSE25097. (C) DEGs in GSE36376. (D) DEGs in GSE76427. Statistically significant DEGs were defined with adjusted $P < 0.05$ and |logFC| > 1 as the threshold value. (E) All the DEGs common to the four datasets are displayed in the Venn diagrams. (F) Up-regulated DEGs common to the four datasets. (G) Down-regulated DEGs common to the four datasets.

that the cell cycle pathway was related to HCC, we performed GSEA on four databases, GSE25097, GSE14520, GSE36376 and GSE76427. The results showed that these data sets had good enrichment score and significance in the cell cycle pathway (Figs. 4E–4H).

## The transcription level of Hub genes in HCC

In order to further verify whether the differentially expressed genes were overexpressed in HCC patients, the ONCOMINE database was used to compare the expression of those genes in tumor tissues and normal tissues. The analysis results showed that these 10 hub genes are overexpressed in liver cancer (Fig. 5A). Then, we used GEPIA2 to draw Box Plots to visualize the different mRNA expression levels of 10 hub genes between cancer samples and normal samples. The result revealed that the expression levels of these 10 genes in HCC tumor samples were higher than in normal samples (Figs. 5B–5K).

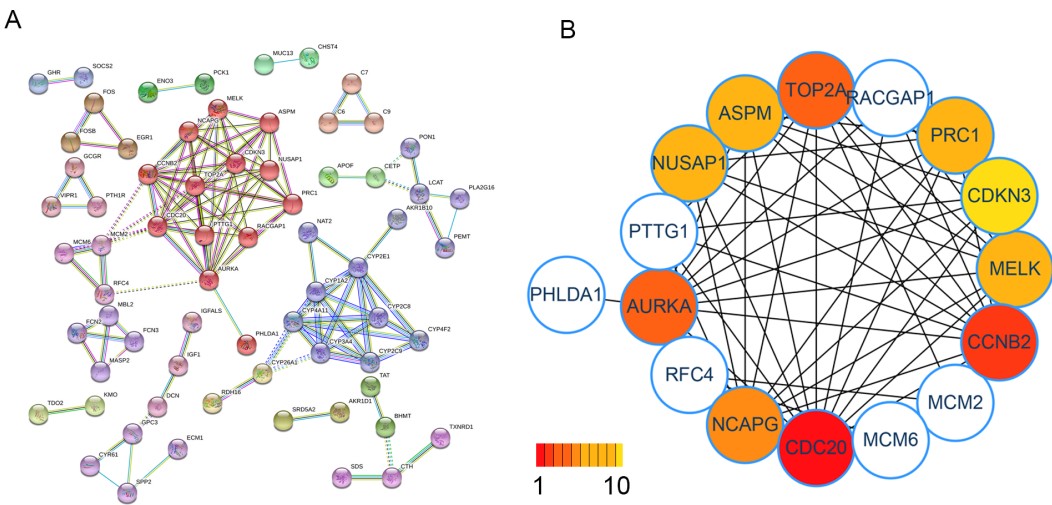

**Figure 3** **Prediction and identification of hub genes in HCC.** (A) A total of 142 DEGs were filtered into the PPI interaction network using the STRING online tools. Red nodes were first cluster which was created using the MCL clustering algorithm. (B) The top-10 hub genes ranked by the MCC algorithm of Cytohubba.

## Establishment of the five-gene-based prognostic gene signature

We used the results of survival analysis to discover the prognostic value of hub genes respectively. The results showed that the OS, PFS and RFS of HCC samples with high expression of those single genes were worse than those with low expression (Fig. 6) (Table 2). Next, the HCC data set of cBioPortal online platform (TCGA, firehose legacy) was applied to get prognostic information of 10 hub genes. The results point out that there is no significant correlation between the hub genes alteration and decreased OS (Fig. 7A). So, we develop a five-gene prognostic signature by LASSO Cox regression analysis. The five genes identified were Cell Division Cycle 20 (CDC20), Non-SMC Condensin I Complex Subunit G (NCAPG), Cyclin B2 (CCNB2), Assembly Factor for Spindle Microtubules (ASPM) and Nucleolus and Spindle Associated Protein 1 (NUSAP1). Then, we validated the prognostic significance of the prognostic signature in HCC patients by the online cBioPortal platform. The results showed that 5 genes were altered in 93 (25%) of 372 samples (Fig. 7C) and those genes alterations were significantly associated with decreased OS (Fig. 7B).

We also calculated the five-gene based risk score for each patient from the TCGA-LIHC cohort and the GSE14520 cohort. The risk score = 0.12 * ExpressionCDC20 + (−0.082) * ExpressionCCNB2 +0.039 * ExpressionNCAPG +0.014 * ExpressionASPM + (−0.04) * ExpressionNUSAP1. The Survminer R package was performed to find the optimal cut-off the risk score. Patients in the high-risk score group shown significantly poorer OS than ones in the low-risk score group (from TCGA $p < 0.001$ and from GSE14520 $p = 0.0003$) (Figs. 7D, 7E). Our results indicated a good performance of the five-gene signature for survival predication.

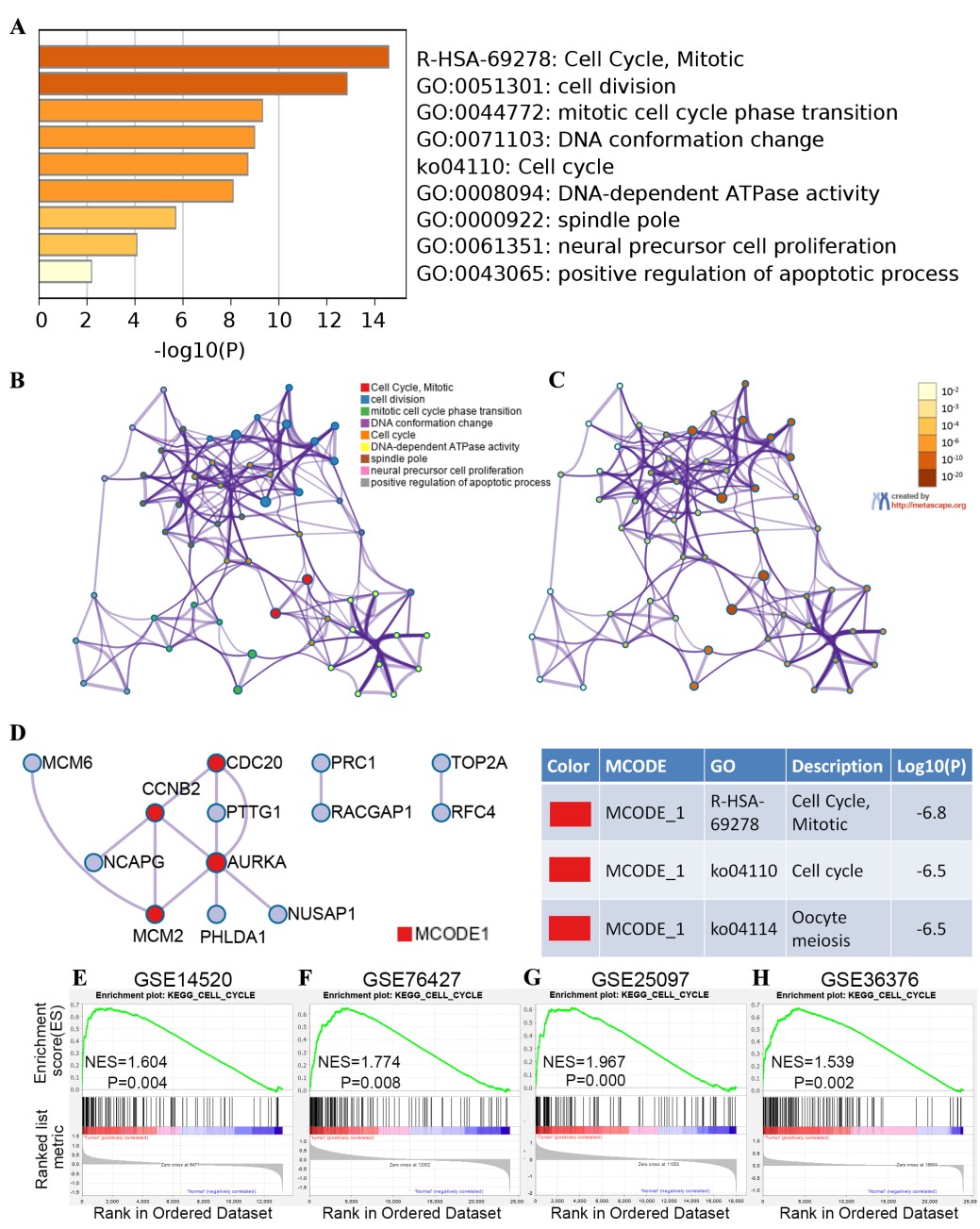

**Figure 4** **Gene Ontology (GO) and Kyoto Encyclopedia of Genes and Genomes (KEGG) analysis of the hub genes.** (A) Significant enrichment of GO annotation and KEGG pathway of hub genes in hepatocellular carcinoma by Metascape ($P < 0.05$). (B) Network of enriched terms: colored by cluster ID, where nodes that share the same cluster ID are typically close to each other. (C) Network of enriched terms: colored by $p$-value, where terms containing more genes tend to have a more significant $p$-value. (D) Protein-protein interaction network and Molecular Complex Detection (MCODE) components identified in the gene lists. (E) Visualization of GSEA results for cell cycles in GSE14520. (F) Visualization of GSEA results for cell cycles in GSE25097. (G) Visualization of GSEA results for cell cycles in GSE36376. (H) Visualization of GSEA results for cell cycles in GSE76427. NES, normalized enrichment score; FDR, adjusted $p$ value.

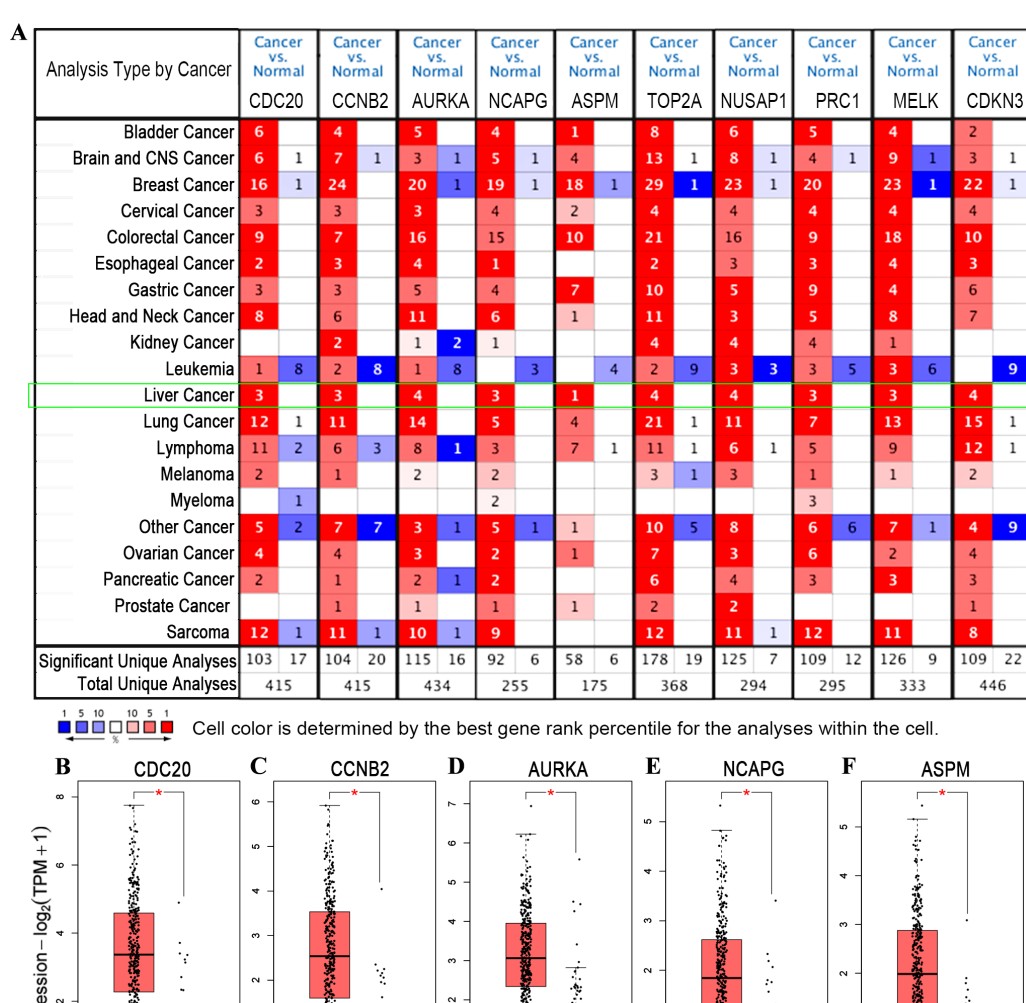

**Figure 5** **Differential analysis of transcription levels of hub genes in hepatocellular carcinoma.** (A) The transcription levels of hub genes in different types of cancers. The graphs showed the numbers of datasets with statistically significant mRNA over-expression (red) or down-expression (blue) of these genes. Differences in the level of transcription of hub genes are displayed in the box plot (GEPIA) which derived from gene expression data for GEPIA comparing the expression of hub genes in HCC tissue ($n = 369$, pink) and normal tissues ($n = 160$, gray). Including (B) CDC20, (C) CCNB2, (D) AURKA, (E) NCAPG, (F) ASPM, (G) TOP2A, (H) NUSAP1, (I) PRC1, (J) MELK and (K) CDKN3. An asterisk (*) shows that $p$ value is less than 0.01.

**Table 2  Kaplan–Meier analysis for ten hub genes in HCC.**

| Gene Symbol | OS(months) | | | RFS(months) | | | PFS(months) | | |
|---|---|---|---|---|---|---|---|---|---|
| | Low | High | Logrank P | Low | High | Logrank P | Low | High | Logrank P |
| CDC20 | 81.9 | 30 | 5.1e−7 | 36.1 | 13.27 | 0.0006 | 33 | 13.27 | 0.0001 |
| CCNB2 | 71 | 46.6 | 0.0013 | 36.1 | 16.73 | 0.0069 | 33 | 13.83 | 0.0011 |
| AURKA | 71 | 37.8 | 0.0011 | 40.97 | 15.63 | 0.0002 | 30.4 | 12.87 | 0.0003 |
| NCAPG | 70.5 | 25.2 | 8.8e−6 | 34.4 | 11.67 | 0.0006 | 29.73 | 10.4 | 0.0002 |
| ASPM | 71 | 45.7 | 0.0002 | 33 | 13.27 | 0.0031 | 36.27 | 15.83 | 0.0002 |
| TOP2A | 71 | 30 | 0.0001 | 36.1 | 11.83 | 0.0001 | 30.4 | 11.33 | 3.0e−6 |
| NUSAP1 | 70.5 | 46.6 | 0.0046 | 36.1 | 13.27 | 0.0010 | 30.4 | 13.33 | 0.0003 |
| PRC1 | 71 | 38.3 | 0.0002 | 36.1 | 12.87 | 0.0005 | 34.4 | 13.27 | 2.3e−5 |
| MELK | 81.9 | 42.4 | 3.7e−5 | 37.23 | 12.87 | 2.5e−5 | 30.4 | 11.6 | 9.4e−6 |
| CDKN3 | 71 | 49.7 | 0.0066 | 30.4 | 17.9 | 0.0219 | 27.6 | 11.83 | 0.004 |

## DISCUSSION

The outcome of HCC patients is not only determined by tumor stage, tumor size, serum markers and liver function, but also closely related to some gene's expression in tumor tissue (*Huitzil-Melendez et al., 2010*). Several researchers have focused on the potential role of gene-signatures based on aberrant mRNA in prognosis prediction of HCC (*Cao et al., 2020*; *Long et al., 2018*; *Liu et al., 2018*; *Wang et al., 2018*; *Kong et al., 2019*). In this study, we screened out 10 hub genes by constructing a PPI network and using cytohubba. Enrichment analysis revealed that most of them are involved in cell cycle progression and survival analysis indicates that HCC patients with the abnormal expression of these genes showed poor OS and PFS. We established a 5-gene signature (including CCNB2, CDC20, NUSAP1, ASPM, and NCAPG) for HCC prognosis prediction. The prognosis predictive performance of the signature was good not only in the TCGA HCC cohort but also in the GSE14520 cohort. All these results indicated that the risk model developed from the five genes could be a useful indicator for HCC survival and to supplement the gap of the clinical prognostic signature of HCC.

The enrichment analysis yielded that the most significant enrichment term was the cell cycle and mitosis. CCNB2 and CDC20 participated in the mitotic cell cycle process. CCNB2, CDC20, NUSAP1, ASPM, and NCAPG involved in cell division. There are reports that CCNB2 is highly expressed in HCC (*Li et al., 2019*). CCNB2 is involved in the development of HCC, which may be a prognostic factor (*Li et al., 2019*). Regulatory protein encoded by CDC20 plays an important role in the occurrence and development of a variety of tumors, which may be related to its participation in the function of anaphase-promoting complex/cyclosome (APC/C) interaction in the cell cycle. Meanwhile, the increase in CDC20 expression has also been shown to be related to the occurrence and development of HCC (*Li et al., 2014*; *Liu et al., 2015*). Because these evidences show that CCNB2 and CDC20 are directly related to the occurrence and development of HCC, we believe that they may play the role of initiator and promoter in the process of HCC. NUSAP1 is one of the most critical microtubule and chromatin binding proteins, which can play a role

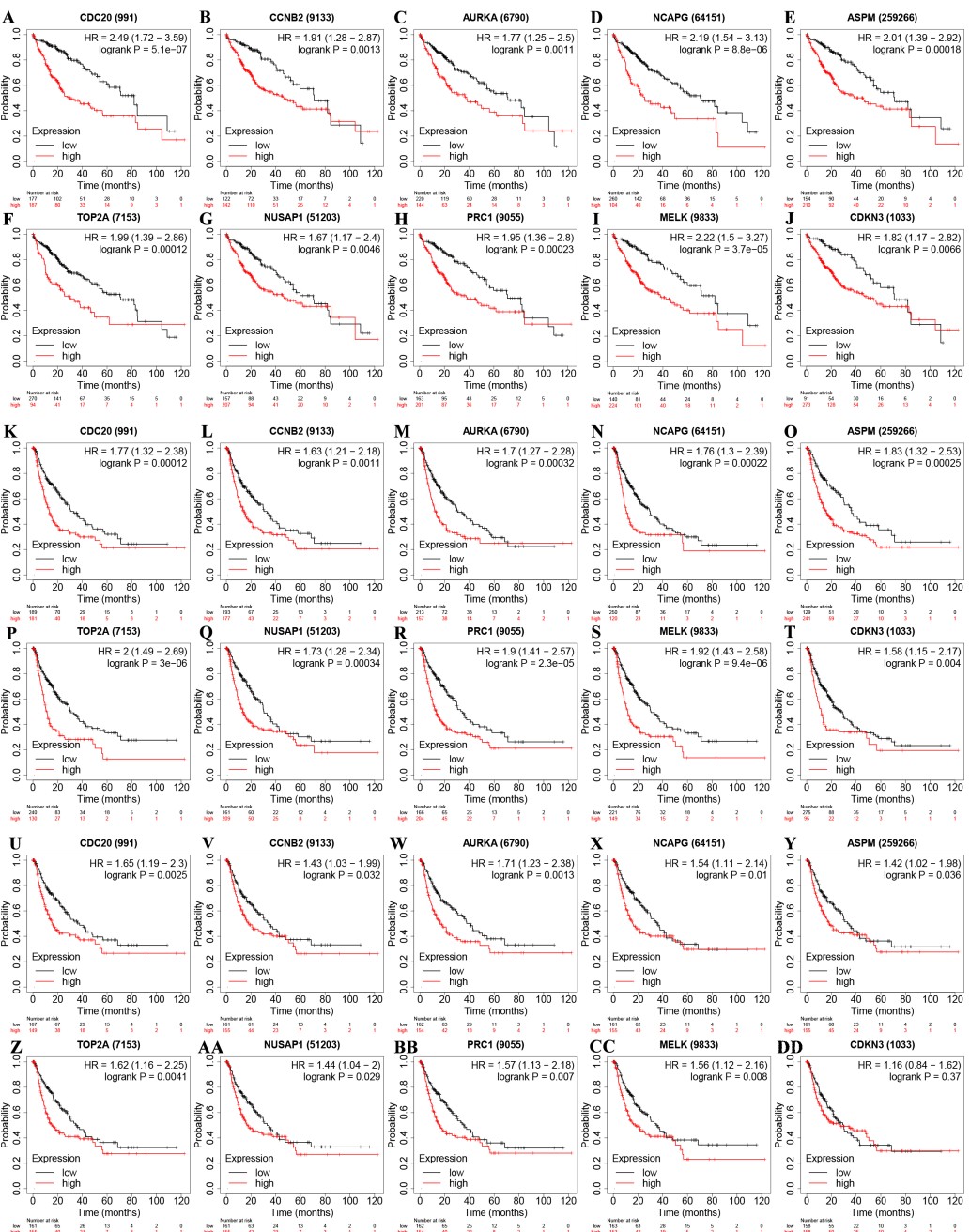

**Figure 6** **Kaplan–Meier analysis for ten hub genes in HCC.** Figure 6 is divided into three parts. First, overall survival (OS) curves of ten hub genes, including (A) CDC20, (B) CCNB2, (C) AURKA, (D) NCAPG, (E) ASPM, (F) TOP2A, (G) NUSAP1, (H) PRC1, (I) MELK and (J) CDKN3. Second, Progression-free survival (PFS) curves of (K) CDC20, (L) CCNB2, (M) AURKA, (N) NCAPG, (O) ASPM, (P) TOP2A, (Q) NUSAP1, (R) PRC1, (S) MELK and (T) CDKN3. Third, Relapse-free survival (RFS) curves of (U) CDC20, (V) CCNB2, (W) AURKA, (X) NCAPG, (Y) ASPM, (Z) TOP2A, (AA) NUSAP1, (BB) PRC1, (CC) MELK and (DD) CDKN3.

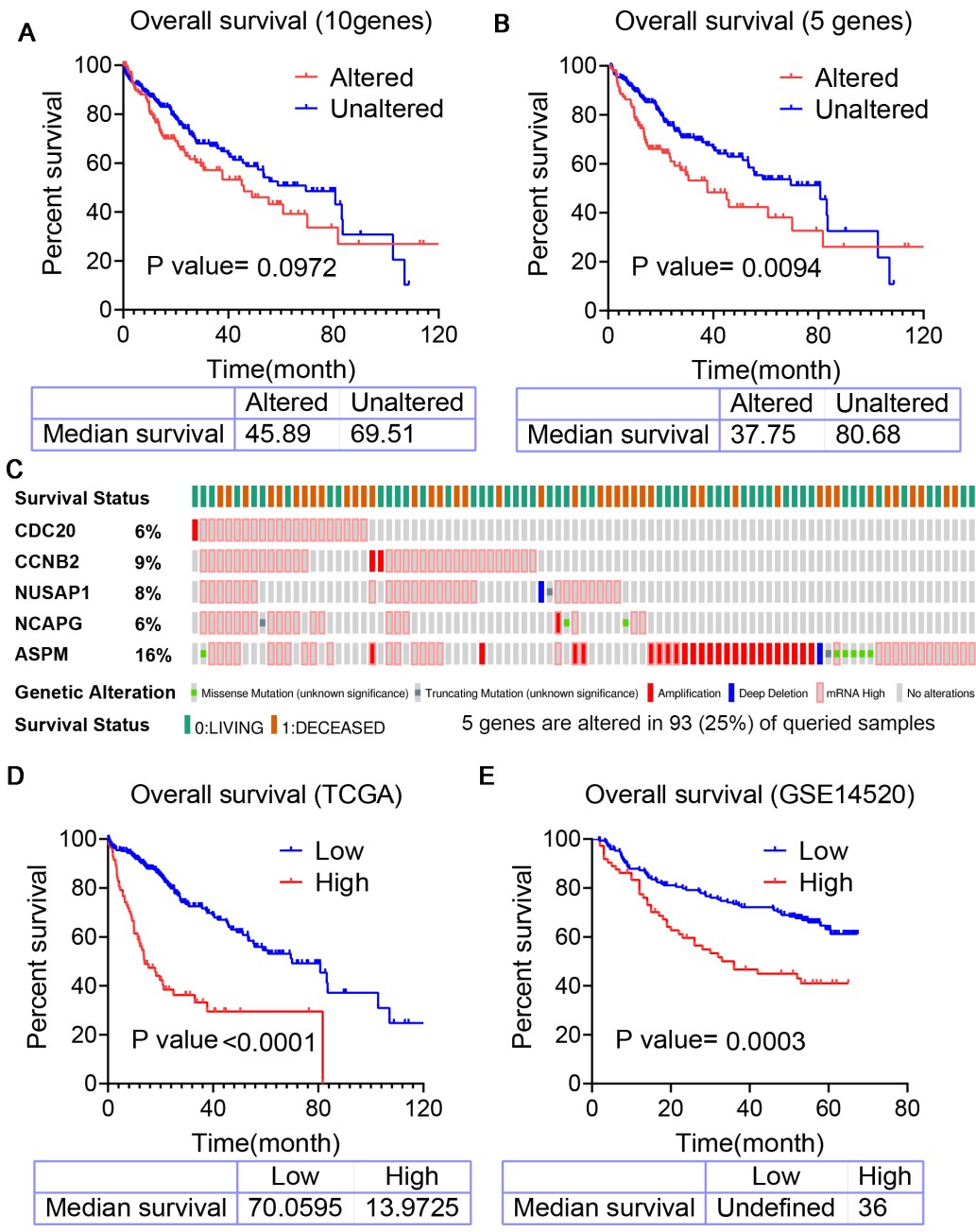

**Figure 7 Kaplan–Meier analysis predicting overall survival for patients with HCC.** (A) Overall survival (OS) curves of ten hub genes in TCGA-LIHC cohort. (B) Overall survival (OS) curves of five-gene signature in TCGA-LIHC cohort. (C) The expression alteration profiles of the six genes in the TCGA-LIHC cohort. (D) Overall survival (OS) curves of five-gene signature compared the survival difference between the high-score and low-score group in TCGA-LIHC cohort. (E) Overall survival (OS) curves of five-gene signature compared the difference between the high-score and low-score group in the GSE14520 cohort.

during mitosis to crosslink microtubules. NUSAP1 expression levels in HCC tissues were higher than those in the adjacent tissues. With the high expression of NUSAP1 in HCC patients, the survival time of the patients also showed a significant decreasing trend (*Wang et al., 2019b*). Unlike CCNB2 and CDC20, although NUSAP1 is highly expressed in HCC tissues with abnormal cell division, we have no direct evidence to prove its effect on the HCC process. Therefore, we believe that the highly expressed NUSAP1 may be temporarily used as a hepatocellular carcinoma Signs of disease. Increased expression of ASPM has been found in HCC. In addition, the expression level of ASPM has an important impact on the biological behavior of cancer cells or the prognosis of patients. ASPM overexpression is a molecular marker predicting poor prognosis (*Lin et al., 2008*). It has been reported that knocking out NCAPG can induce the division of HCC cells and even inhibit its deterioration in an in vitro environment. In contrast, the overexpression of NCAPG is also related to the recurrence of HCC patients (*Zhang et al., 2018*). Studies have shown that the prognostic effect of NCAPG makes it a new biomarker for predicting whether recurrence will occur after surgical removal of the tumor (*Wang et al., 2019a*). These evidences indicate that ASPM and NCAPG may be closely related to the prognosis of HCC patients, including positive, worsening or recurring prognostic results. Inhibiting the proliferation of HCC by targeted drugs to cause cancer cell apoptosis and curing cancer has become a new method of current cancer clinical treatment, such as Compound Kushen, which inhibits the cell cycle (*Feitelson et al., 2015*; *Cui et al., 2019*). However, it not only provides new biomarkers for the target treatment of HCC, but also provides a new plan for the clinical management of HCC.

In summary, consistent with our results, these 5 genes are all related to the prognosis of liver cancer. To our knowledge, the prognostic model associated with the five-gene signature may be a useful prognostic tool for liver cancer clinically. The risk score can be based on the mRNA expression levels of the five prognostic genes. In clinical practice, it may be more routine and cost-effective for all HCC patients. However, some limitations of our research should be considered. Firstly, we need to use more databases to verify the accuracy of this model. When we validated the five-gene signature by GSE14520 cohort, the median survival time of low-risk score group was undefined. It indicated that the follow-up time was not long enough in an original study. Secondly, at present, there are several reports that have screened out different genes signature, which needed to have a better and more accurate method to verify the effectiveness. Thirdly, expression profiling can only detect the change of gene expression level in our study, subsequent experiments are required for providing the information of those protein expression levels. Finally, we lack the molecular mechanisms of interaction between these genes, and we will incorporate these for further exploration.

## CONCLUSIONS

This study aims to perform a comprehensive bioinformatics analysis of DEGs in four hepatocellular carcinoma datasets to discover potential biomarkers and predict their clinical effects. We established a five-gene signature (CCNB2, CDC20, NUSAP1, ASPM,

and NCAPG) to predict overall survival of HCC, which may contribute to the clinical decision-making of HCC treatment for different individuals.

# ACKNOWLEDGEMENTS

The authors are grateful to all patients who provided samples to the public databases.

## Funding

The present study was funded by the Provincial Medicine and Health Science Technology Development Program Shandong (NOs. 2017WS822 and 2017WS558) and the Innovation and Entrepreneurship Training Program for College Students (NO. 201910440001). The funders had no role in study design, data collection and analysis, decision to publish, or preparation of the manuscript.

## Grant Disclosures

The following grant information was disclosed by the authors:
Provincial Medicine and Health Science Technology Development Program Shandong: 2017WS822, 2017WS558.
Innovation and Entrepreneurship Training Program for College Students: 201910440001.

## Competing Interests

The authors declare there are no competing interests.

## Author Contributions

- Lei Yang and Hongxing Li conceived and designed the experiments, performed the experiments, analyzed the data, prepared figures and/or tables, authored or reviewed drafts of the paper, and approved the final draft.
- Weilong Yin conceived and designed the experiments, performed the experiments, prepared figures and/or tables, authored or reviewed drafts of the paper, and approved the final draft.
- Xuechen Liu conceived and designed the experiments, prepared figures and/or tables, authored or reviewed drafts of the paper, and approved the final draft.
- Fangcun Li performed the experiments, authored or reviewed drafts of the paper, and approved the final draft.
- Li Ma analyzed the data, authored or reviewed drafts of the paper, and approved the final draft.
- Dong Wang conceived and designed the experiments, performed the experiments, analyzed the data, prepared figures and/or tables, authored or reviewed drafts of the paper, teaching of bioinformatics methods, and approved the final draft.

## Data Availability

The raw measurements are available in the Supplementary Files.

## Supplemental Information

Supplemental information for this article can be found online at http://dx.doi.org/10.7717/peerj.11273#supplemental-information.

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
