# Peer review of "Identification of a five-gene signature in association with overall survival for hepatocellular carcinoma"

_PeerJ, doi:10.7717/peerj.11273_

## Round 0.1 · original submission · Minor Revisions

Please address all reviewer comments and feedback.

·

Basic reporting

The study was carried out with detailed bioinformatics approach. Reasonable references were used to cover all the background about the research. The figures and tables are clear to understand the goal of the study. Also the results are focused.
Few points would be important to add,
1. Are the normal tissues collected from the adjacent part from of HCC patients?
2. It would be important to know if HCC patients had any other health conditions.

Experimental design

Experimental designs are hypothesis directed and answered most of the necessary questions. Also, the experimental approaches are reasonable and precise. The methods are explained in full details.

Validity of the findings

The findings are interesting and this study has provided sufficient data to strengthen their hypothesis.

Additional comments

Very detailed approach.

Reviewer 2 ·

Basic reporting

The writing is well organized and professional

Experimental design

The details of the analysis is well described

Validity of the findings

The methods are well described and that support their findings of the targets at the RNA level. However, a few more additional experiments will enhance the quality of the paper.

Additional comments

This article aims to identify markers for overall survival in HCC. Overall the methods are well detailed and the analysis well presented. The authors have identified some of the shortcomings such as validation of protein. Some of my comments are as follows

1. Line 178 - 5 genes were altered in 93 (25%) of 372 samples. This appears to be a small percentage and sample size to be a marker without any protein verification. Any verification at the protein level is important and will increase the impact of the paper,
2. Discussion lines 199 to 220 – For most targets, the authors have mentioned how they might have an important role but does not describe the role in details. Kindly elaborate the discussion further.

Reviewer 3 ·

Basic reporting

This is a well-written manuscript that identified a prognostic model with 5 genes by transcriptomic analysis on hepatocellular carcinoma databases. This 5 gene-based risk model was further validated by TCGA-LIUC and GSE14550 databases.

Experimental design

Some comments:
1. In the introduction, the authors need to introduce the current gap of the prognosis model for liver cancers and how this study can fill the gap.
2. Figure 1, I would suggest the authors to include a supplementary table listing the 15% upregulated genes in the tumor tissues. This would help increase the impact by allowing further exploration of the signatures by readers.
3. Are these 5 genes relevant to the clinical-pathological characteristics of the disease such as TNM stage, pathological type. etc?

Validity of the findings

The results have some promise but given the contradictory nature can you back it up with some experimentation to validate the finding?

Additional comments

No.

Reviewer 4 ·

Basic reporting

There are several grammatical mistakes at different places in the manuscript. For example in the methods part of the highlight section in the beginning there is a sentence that ends abruptly (" Then, gene set enrichment analysis (GSEA) was used to identify. "). There are several such mistakes and the authors should carefully review and correct these grammatical errors.

The literature review provided by the authors is very brief and almost irrelevant. Readers and reviewers would definitely gain from knowing other research efforts to identify prognostic gene expression based biomarkers in HCC or even other types of cancers.

Experimental design

The experiment design and the analysis performed by the authors was thorough. However, it would be beneficial if they compared their work and results with other efforts to identify gene-expression based biomarkers.

Validity of the findings

The authors showed convincingly that their 5-gene signature was sufficiently predictive of OS and PFS.

Additional comments

The analysis was straightforward and easy to follow. The authors clearly show the prognostic power of the 5-gene expression based signature in HCC. Some literature review and relevant to the point motivation for the study will help readers appreciate the work.

Reviewer 5 ·

Basic reporting

Manuscript details
Journal: Peerj
Manuscript ID: Peerj - 55265
Type of manuscript: Article
Title: Identification of a five-gene signature in association with overall survival for hepatocellular carcinoma
In this manuscript entitled “Identification of a five-gene signature in association with overall survival for hepatocellular carcinoma”. There is some useful information, however, the further modification is required, and I have provided several advices shown below.
1. This article uses a lot of data analysis, but the main reason for selecting these dataset is not very clear. I hope the author can discuss it in detail.
2. The data has been analyzed in many aspects. Due to the wide range of genetic influences, it is recommended to use genetically modified mice for further research.
3. The authors only discussed the mortality of genes. I suggested that related drug development strategies should also be discussed together
To sum up, the manuscript should be minor revision.

Experimental design

Reasonable experiment design

Validity of the findings

Manuscript details
Journal: Peerj
Manuscript ID: Peerj - 55265
Type of manuscript: Article
Title: Identification of a five-gene signature in association with overall survival for hepatocellular carcinoma
In this manuscript entitled “Identification of a five-gene signature in association with overall survival for hepatocellular carcinoma”. There is some useful information, however, the further modification is required, and I have provided several advices shown below.
1. This article uses a lot of data analysis, but the main reason for selecting these dataset is not very clear. I hope the author can discuss it in detail.
2. The data has been analyzed in many aspects. Due to the wide range of genetic influences, it is recommended to use genetically modified mice for further research.
3. The authors only discussed the mortality of genes. I suggested that related drug development strategies should also be discussed together
To sum up, the manuscript should be minor revision.

Additional comments

Manuscript details
Journal: Peerj
Manuscript ID: Peerj - 55265
Type of manuscript: Article
Title: Identification of a five-gene signature in association with overall survival for hepatocellular carcinoma
In this manuscript entitled “Identification of a five-gene signature in association with overall survival for hepatocellular carcinoma”. There is some useful information, however, the further modification is required, and I have provided several advices shown below.
1. This article uses a lot of data analysis, but the main reason for selecting these dataset is not very clear. I hope the author can discuss it in detail.
2. The data has been analyzed in many aspects. Due to the wide range of genetic influences, it is recommended to use genetically modified mice for further research.
3. The authors only discussed the mortality of genes. I suggested that related drug development strategies should also be discussed together
To sum up, the manuscript should be minor revision.

---

## Round 0.2 · accepted · Accept

The manuscript is being accepted for publication since the authors have addressed all comments from reviewers.

·

Basic reporting

Grammar need to corrected in line 35. "up-regulation" should be "up-regulated".

Experimental design

No comment

Validity of the findings

No comment

Additional comments

Very detailed and covered all the queries.

Reviewer 3 ·

Basic reporting

All the comments were properly addressed.

Experimental design

N/A

Validity of the findings

N/A

Additional comments

Thank you for properly addressing the comments!

Reviewer 4 ·

Basic reporting

The revision looks good.

Experimental design

The revision looks good.

Validity of the findings

The revision looks good.

Additional comments

The revision looks good.

Reviewer 5 ·

Basic reporting

The manuscript is suitable to be published.

Experimental design

The manuscript is suitable to be published.

Validity of the findings

The manuscript is suitable to be published.

Additional comments

The manuscript is suitable to be published.